

# Spatiotemporal Variation of Van der Burgh's coefficient in a salt plug estuary

Dinesh Chandra Shaha[1,2], Yang-Ki Cho[*,2], Bong Guk Kim[2], Md. Rafi Afruz Sony [1], Sampa Rani Kundu[3] Md.

5      Faruqul Islam[4]

[1]Department of Fisheries Management, Bangabandhu Sheikh Mujibur Rahman Agricultural University, Gazipur

1706, Bangladesh

10     [2]School of Earth and Environmental Science/Research Institute of Oceanography, Seoul National University,

Seoul, Korea

[3]Deparment of Oceanography, Chonnam National University, Gwangju, Korea
[4]Hydrography Division, Mongla Port Authority, Bagherhat, Bangladesh.

Corresponding author[*]

Phone: (02) 880-6749

Fax: (02) 9205333

20     Email: choyk@snu.ac.kr



**Abstract**

 Saltwater intrusion in estuaries is expected to become a more serious issue around the world due to climate change. Van der Burgh's coefficient, *K,* is a good proxy for describing the relative contribution of the tide-driven and gravitational components of salt transport in estuaries. However, debate continues over the use of *K* value for an estuary where *K* should be constant or spatially varying or a time-independent factor for different river discharge conditions. In addition, whether *K* functions in an inverse salinity gradient area of a salt plug estuary has not been examined thus far.  In this study, we determined *K* during spring and neap tides in the dry (<30 m$^{-3}$s$^{-1}$) and wet (>750 m$^{-3}$s$^{-1}$) seasons in a salt plug estuary with an exponentially varying width and depth to examine the relative contributions of tidal versus density-driven salt transport mechanisms. High-resolution salinity data were used to determine *K*. Gravitational circulation (*K*~0.8) was entirely dominant over tidal dispersion during spring and neap tides in the wet season such that salt transport upstream was effectively reduced, resulting in the estuary remaining in a relatively fresh state. In contrast, during the dry season, *K* increases gradually seaward and landward (*K*~0.74) from the salt plug area (*K*~0.65), similar to an inverse and positive estuary, respectively. As a result, density-induced inverse gravitational circulation between the salt plug and the sea facilitates inverse estuarine circulation. On the other hand, positive estuarine circulation between the salt plug and the river area arose due to density-induced positive gravitational circulation induced by the tide during the dry season, causing the intrusion of high-salinity bottom water upstream. Our results explicitly show that *K* varies spatially and depends on the river discharge. This result provides a better understanding of the distribution of hydrographic properties as well as the distributions of pollutants, nutrients and biota within large estuaries.

*Keywords*: Van der Burgh's coefficient, salt transport, spring-neap tides, salt plug estuary, river discharge



## 1. **Introduction**

A quantitative understanding of the characteristics of salinity distribution and transport under various

environmental conditions is essential for the interpretation of an estuary's physical, chemical, biological, and

ecological status. Salt water intrusion into tropical estuaries has received a substantial attention in recent years

due to changes in rain frequency and intensity levels. In addition, salt water intrusion can be aggravated by

decreasing river discharge resulting from upstream barrages required for drinking and irrigation water (Shaha and

Cho, 2016). Changes in river discharge levels alter estuarine circulation, stratification, flushing times, salt water

intrusion and the transport of biota and dissolved and particulate materials such as salt, pollutants, nutrients and

organic matter (Azevedo et al., 2010; Lee and An, 2015; Savenije, 2012; Shaha and Cho, 2016; Valle-Levinson,

2010). Therefore, it is particularly important to understand the responses of estuarine salt transport mechanisms

to temporal changes in river discharge levels because salt water intrusion may lead to shortages of drinking and

irrigation water (Khan et al., 2011), decreased rice production (Mirza, 2004), reduced freshwater fish habitat

(Dasgupta et al., 2014) and inadequate industrial fresh water supplies (Mirza, 1998).

The relative weights of horizontal salt transport mechanisms, both the tidal and density-driven dispersion types,

in estuaries can be expressed using Van der Burgh's coefficient, $K$ (Savenije, 2005; Shaha and Cho, 2011; Van

der Burgh, 1972). Tidal mixing and density-driven mixing vary along the axis of an estuary according to the tidal

influence and the volume of river discharge. Tide-driven mixing usually dominates downstream; a combination

of tidal and gravitational components influences the central regimes, whereas gravitational mixing tends to

dominate upstream (Shaha et al., 2010). Therefore, a constant $K$ value for an estuary, as suggested in earlier work

(Gisen, 2015; Savenije, 1993, 2005; Zhang and Savenije, 2016), can not accurately represent the nature of salt

transport in estuaries for high and low river discharge conditions. Shaha and Cho (2011), who suggested a



modified equation to account for the exponential variation in estuarine widths, examined the spatial variability of

$K$ along the axis of a small, narrow estuary.

Nonetheless, debate continues regarding the use of $K$ for an estuary, i.e., whether this value should be constant or

spatially varying and/or whether it can serve as a time-independent factor for varying river discharges and

geometries. $K$ is assumed to be a time-independent parameter, and every estuary has its own characteristic $K$

value (Savenije, 1986; Savenije, 1993, 2005). Shaha and Cho (2011) found that $K$ values not only vary along an

estuary owing to different salt transport mechanisms but also depend on river discharge levels in their test of a

small, narrow estuary. In contrast, Gisen (2015) assumed $K$ to be independent of the river discharge level, finding

instead that it depends on the topography. Conversely, Zhang and Savenije (2016) suggested a constant $K$ value if

the depth is constant along the estuary. However, the depth typically varies along an estuary. For instance, earlier

research showed that the depth varied in 15 estuaries among 18, and was constant only in three (Maputo, Thames

and Tha Chin) (Zhang and Savenije, 2016). In the present study, we focus on determining $K$ during spring and

neap tides in the dry ($<30$ m$^{-3}$s$^{-1}$) and wet ($>750$ m$^{-3}$s$^{-1}$) seasons in a salt plug estuary with an exponentially

varying width and depth to examine the relative contributions of tidal versus density-driven salt transport

mechanisms. In addition, whether $K$ functions in an inverse salinity gradient area of a salt plug estuary has not

been examined thus far.  Therefore, we examined as well whether $K$ can serve in an inverse salinity gradient of

such a salt plug estuary in this study.

The Pasur River Estuary (PRE) is the longest ($>164$ km) estuary in the south western part of the Ganges-

Brahmaputra delta into Bangladesh. Salt water intrusion in the PRE has received substantial attention in recent

years due to increases in the magnitude and frequency of salt water intrusion upstream due to climate change, a

predicted sea-level rise of 30 cm by the year 2050 (IPCC, 2007), and decreases in river discharge levels resulting





from an upstream barrage (Shaha and Cho, 2016). Most previous studies focused primarily on analyzing the

relationship between discharge and salinity in the PRE (Mirza, 1998, 2004; Rahman et al., 2000; Uddin and

Haque, 2010). A few studies of the hydrology of mangrove ecosystems (Wahid et al., 2007), fish biodiversity

(Gain et al., 2008; Gain et al., 2015), surface-water quality (Rahman et al., 2013) and nutrient distributions

(Rahaman et al., 2014) have been conducted in the PRE. Recently, a new type of salt plug formation was

discovered in the multi-channel Pasur River Estuary (PRE); this was found to have been caused by decreasing

river discharges levels resulting from an upstream barrage (Shaha and Cho, 2016). However, earlier work

typically omitted the details of salt transport mechanisms in the PRE, and these details are necessary for a

complete understanding of the hydrodynamics and causes of instances of salt water intrusion upstream.

Therefore, in this study, we applied the equation suggested by Shaha and Cho (2011) to determine $K$ during

spring and neap tides in the dry and wet seasons. We sought to determine the variations in salt transport

mechanisms in the PRE considering its exponentially varying width and depth, and to assess the influence of

river discharge levels on $K$.

## 2.  Material and methods

### 2.1. Study area

There are three distinct seasons in Bangladesh: a dry summer from March to June; a rainy monsoon season from

July to October; and a dry winter from November to February (Rashid, 1991). River discharge is strongly seasonal.

During the wet season (monsoon), approximately 80% to 90% of the annual rainfall occurs. Maximum discharge

occurs between July and October (wet season). In contrast, river discharge is negligible from November to June

(dry season).



The Pasur River is the most commercially important river that experiences salt water intrusion upstream in the southwestern coastal zone of Bangladesh (Fig. 1a). The Pasur River bifurcates into two distributaries, the Shibsa River and the Pasur River, at Akram Point before entering the Bay of Bengal (Fig. 1b). Approximately 68 km

upstream from Akram Point, the Chunkhuri Channel connects the Pasur River to the Shibsa River at Chalna. The interconnecting channel contributes to complex water circulation between the Pasur and Shibsa River estuarine systems (Shaha and Cho, 2016). There is no direct link between the Shibsa River upstream and the major freshwater source, the Ganges. Therefore, high salinization occurs in the Shibsa estuary relative to the PRE in the dry season owing to the lack of freshwater discharge and precipitation (Shaha and Cho, 2016). On the other hand,

the Pasur River is directly connected to the main freshwater source of the Ganges through the Gorai-Madhumati-Nabaganga-Rupsha-Pasur (GMNRP) river system (Fig.1a). The Ganges, originating in the Himalayas and the third largest river (in terms of discharge) in the world, was unregulated prior to the construction of the Farakka barrage in India in 1975. This diversion diminished the average dry season flow in the Ganges from 3114 $m^3$ $sec^{-1}$ during the pre-Farakka period to 2010 $m^3$ $sec^{-1}$ in the post-Farakka period (Islam and Gnauck, 2011; Mirza,

2004). As a result, the dry-season discharge in the Gorai River, the major distributary of the Ganges, was reduced from a pre-Farakka mean flow of 190 $m^3$ $sec^{-1}$ in 1973 (Islam and Gnauck, 2011; Mirza, 2004) to post-Farakka mean flows of 51 $m^3$ $sec^{-1}$ in 1977 and 10 $m^3$ $sec^{-1}$ in 2008 (Islam and Gnauck, 2011). Consequently, salt water intrusion has extended as far as ~164 km (29 March 2014) from the estuarine mouth (at Hiron Point) to a head at Lohagara, Narail, during the spring tide in the dry season (Shaha and Cho, 2016).






## 2.2. Data

The bathymetric chart of the PRE from Harbaria to Chalna used in this study was collected from the Mongla Port Authority. The cross-sectional depths, areas and widths at different sampling stations within the study site are shown in Fig. 2. In addition, river discharge data from January to December of 2014 were collected from a non-tidal discharge station on the Gorai River, the main upstream freshwater source of the PRE. Tidal water level data for Mongla Port and Hiron Point were obtained from the Mongla Port Authority (Fig. 1b). The tidal range varied from 1.6 to 3.0 m at Hiron Point and from 2.2 to 4.0 m at Mongla Port during the neap and spring tides, respectively (BIWTA, 2014). The tidal range is higher at Mongla Port than at Hiron Point.

Nine longitudinal depth profiles of salinity were taken using a conductivity-temperature-depth (CTD) profiler (Model: *In-situ* Aqua TROLL 200, In-situ Inc., Fort Collins, Colorado, USA) along the main axis of the Pasur River from Harbaria to Rupsha Bridge (> 60 km). Speed boats or mechanized boats are not allowed to operate southward from Harbaria to the estuary mouth due to the strong tidal influence. Longitudinal transects were taken at high water levels during both neap and spring tides in the wet and dry seasons from February to December of 2014 (Table 1). The use of a global positioning system (GPS) ensured that precise data was obtained at the sampling stations. The nominal distance between stations was approximately 3 km owing to the low salinity gradient (~ 0.05 km$^{-1}$) along the estuary.

## 2.3. Methods

A one-dimensional salinity model is used to predict the salinity in estuaries (Savenije, 2012). Under a steady-state condition, the salt balance equation (Savenije, 2012) can be written as follows:

$$S_i(x) - S_f = \frac{A(x)}{Q} D_i(x) \frac{\partial S_i}{\partial x} \qquad (1)$$




where $D_i(x)$ is the longitudinal dispersion coefficient, $S_f$ is the freshwater salinity (usually close to zero), $Q$ is the freshwater discharge, $S_i(x)$ is the salinity along the estuary at the high water slack, and $A(x)$ is the cross-sectional area. By combining the salt balance equation with the Van der Burgh equation, the longitudinal variation of the

effective dispersion is given as follows (Savenije, 2005):

$$\frac{\partial[D(x)]}{\partial x} = K(x)\frac{Q}{A(x)} \qquad (2)$$

where $K(x)$ is the dimensionless Van der Burgh coefficient. The effective dispersion decreases upstream, showing a direct proportion against the velocity ($Q/A = U$) of the freshwater discharge (Savenije, 2005; Van der Burgh, 1972).


Van der Burgh's method, related to a decrease in the effective dispersion in the upstream direction, is similar to a number of methods developed by other scientists (Hansen and Rattray, 1965; Ippen and Harleman, 1961; Stigter and Siemons, 1967). Among these methods, the theory of Hansen and Rattray (1965) is most similar to Van der Burgh's method.  Hansen and Rattray (1965) limited their theory to the central zone of a narrow estuary with a

constant cross-section, presuming that the salinity in the central zone would decrease linearly upstream. On the basis of these strong assumptions, the tide-driven horizontal dispersion $D_t$ is given as follows (Savenije, 2005):

$$\frac{\partial D_t(x)}{\partial x} = \frac{Q}{A(x)} \qquad (3)$$

The proportion of the tide-driven dispersion $D_t(= D\partial S/\partial x)$ to the total dispersion $D(= SU_f = SQ/A)$ is termed the

estuarine parameter, $v$ (Savenije, 2005). The estuarine parameter can be used to characterize the nature of salt transport in estuaries. The contribution by the diffusive portion vs the advective portion of the total salt flux into the estuary can be given as a function of $x$:



$$v(x) = \frac{D_t(x)}{D(x)} = \frac{D(x)A(x)}{S_0 Q}\frac{\partial S}{\partial x} \tag{4}$$

The parameter $v$ can fluctuate between 0 and 1 (Valle-Levinson, 2010). Shaha and Cho (2011) found that $v$ decreased from almost unity near the mouth to zero at the end of salt the intrusion curve, indicating a transition from tide-driven to salinity-driven mixing. Shaha and Cho (2011) investigated the variability of $v$ along the axis of the Sumjin River Estuary. In the present study, $v(x)$ was calculated using equation (4). A combination of Eqs. (3) and (4) can be given as follows:

$$\frac{\partial[D(x)]}{\partial x} = \left\{\frac{1}{v(x)} - \frac{D(x)A(x)}{v(x)Q}\frac{\partial[v(x)]}{\partial x}\right\}\frac{Q}{A(x)} \tag{5}$$

Shaha and Cho (2011) showed the relationship between $K(x)$ and $v(x)$ with Eqs. (2) and (5) as follows:

$$K(x) = \frac{1}{v(x)}\left\{1 - \frac{D(x)A(x)}{Q}\frac{\partial[v(x)]}{\partial x}\right\} \tag{6}$$

$D(x) = (QS(x)/A(x))/(\partial S/\partial x)$ is applicable to well-mixed estuaries, but strictly inapplicable to stratified conditions (Dyer, 1997). On the basis of the stratification parameter, the PRE is a well-mixed to partially-mixed estuary during spring and neap tides in the dry and wet seasons. The values of $K$ calculated using Eq. (6) exceed the recommended limit of 1 (Shaha and Cho, 2011).

To limit the feasible range of $0 < K < 1$ in an estuary with an exponentially varying width, an exponential function is considered with the proportion of tidal dispersion to the total dispersion, $\exp(D_t/D)$, following the theory of McCarthy (1993). Shaha and Cho (2011) proposed a spatially varying $K$ value for an exponential shaped estuary, as follows:



$$K(x) = \frac{1}{\exp(v(x))} \left\{ 1 - \frac{D(x)A(x)}{Q} \frac{\partial[v(x)]}{\partial x} \right\} \tag{7}$$

Equation (7) limits the feasible range of $K$, as suggested by several researchers (Eaton, 2007; Savenije, 2005). In

addition, $K$ also describes the spatial variation of the tidal- and density-driven mixing of salt transport in the

small, narrow estuary (Shaha and Cho, 2011). The $K$ value has been scaled on the basis of the $v$ value, and it

ranges from 0 to 1 (Shaha and Cho, 2011). If $K < 0.3$, the total salt transport is driven by diffusive processes (e.g.,

tidal mixing), as in unidirectional net flows. If $K \sim > 0.8$, up-estuary salt transport is controlled by advection, i.e.,

by discharged-induced gravitational circulation. In this case, mixing processes are weak, as in a highly stratified

estuary (Valle-Levinson, 2010). The system exhibits contributions from advective and diffusive processes to the

upstream salt transport, if $0.3 < K < 0.8$. If $0.51 < K < 0.66$, the dispersion is proportional to the salinity gradient, meaning

it is driven by the longitudinal density gradient (Zhang and Savenije, 2016).


## 3. Results and discussion

### 3.1. Longitudinal salinity distribution

Longitudinal sections of salinity were taken during spring and neap tides in the dry and wet seasons along the

main axis of the PRE from Harbaria to Rupsha Bridge (Figs. 3). A salt plug formed near Chalna in the PRE, 68

km upstream from the estuary mouth (Akram Point), owing to export of salt water from the Shibsa river estuary

through the Chunkhuri Channel during the dry season (Fig. 3). The salinity is reduced gradually landward (from

Chalna to Rupsha Bridge) and seaward (from Chalna to Harbaria) from the salt plug area, similar to a positive

estuary and an inverse estuary, respectively (Shaha and Cho, 2016; Valle-Levinson, 2010; Wolanski, 1986). The

salt plug existed from December to June in the PRE, and isolated the upper reaches of the estuary from the



coastal water. In contrast, during the wet season, the salt plug advected to the Bay of Bengal and created a typical

estuarine condition in which salinity decreases with an increase in the distance upstream, moving away from the

mouth.

The depth-averaged salinity ranged between 10 and 17.8 and did not vary significantly between spring and neap

tides in the dry season (Figs. 4a-b). In contrast, during the wet season, the salinity varied between 0.15 and 3.0

(Figs. 4a and c). The salinity was lower during neap tides than during spring tides in the wet season, most likely

due to weaker turbulence during neap tides as well as higher river discharge levels, which act in concert to

enhance gravitational circulation and thus decrease the salinity level. Moreover, strong tidal currents during

spring tides tend to suppress gravitational circulation (Geyer, 1993; Savenije, 2005) and thus increase salinity.


## 3.2 Spatial variation of Van der Burgh's coefficient during the wet season

Van der Burgh's coefficient characterizes estuarine salt flux mechanisms which includes both tide-driven and

gravitational circulation (Savenije, 2006). Gravitational circulation is driven by river discharge and density

gradients. If the mixing is mostly of the density-driven type, the dispersion should then be proportional to the

salinity gradient (Savenije, 2005; Zhang and Savenije, 2016). By contrast, if the mixing is mostly the tide-driven

form, then the dispersion is essentially constant. In reality, there is a combination of both mechanisms, whereby

tidal mixing is prominent near the mouth of the estuary and gravitational mixing is influential further upstream,

where the salinity gradient is steep (Savenije, 2005).

Van der Burgh's coefficient was calculated using Eq. (7) along the length of the PRE, from Harbaria to Rupsha

Bridge, using the depth-averaged salinity and the available bathymetric information. Figure 5a depicts the spatial

variation of Van der Burgh's coefficient from Harbaria to Rupsha Bridge in the dry and wet seasons. Landward



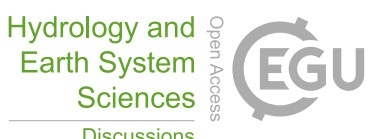

(up to 10 km from Harbaria), where $K$ ranged from 0.7 to 0.8, the transport of salt was dominated by both tide-driven and gravitational circulation during the wet season. Discharge-driven gravitational circulation dominated

over tidal dispersion during the wet season and reduced the transport of salt in this area. The combined influence of tide-driven and gravitational circulation determined the salt transport in this area due to a 20% increase of the $M_2$ tidal amplitude from Hiron Point to Mongla Port. Upstream (over 10 km from Harbaria), where $K > 0.8$, discharge-driven gravitational circulation weakened salt transport during the wet season due to high river discharge levels ($>750$ $m^3s^{-1}$). Gravitational circulation was stronger during neap tides than it was during spring

tides in the wet season, as during neap tides, weaker turbulence and higher river discharge levels combined to enhance gravitational circulation. Consequently, the PRE became much less saline (fresher) during neap tides than during spring tides (Fig. 4c). Therefore, the circulation in the PRE during the wet season resembled that of a typical estuary.

Additionally, this result clearly shows the effects of the basin's morphology (here, the estuarine length) on salt transport during the wet season. In the PRE (a long estuary), discharge-driven gravitational circulation lessened salt transport substantially in the central regimes, whereas the combined influence of tide-driven and gravitational circulation was found to determine salt transport in the central regimes of a small estuary due to the intense tidal influence (Shaha and Cho, 2011).


**3.3 Spatial variation of Van der Burgh's coefficient during the dry season**

Salt transport mechanisms did not vary significantly between spring and neap tides (Fig. 5b) during the dry season, when the river discharge was low ($<30$ $m^3s^{-1}$). During the dry season, the spatial variation of $K$ indicates a gradual rise seaward of the salt plug (from Chalna to Harbaria, Fig. 5), similar to an inverse estuary. The $K$ value

of ~ 0.65 near Chalna suggests density-driven inverse gravitational circulation between Chalna and Harbaria as



the *K* value was reduced to 0.65 from 0.74 (Figs. 5-6). This inverse gravitational circulation results from the adjustment of the density gradient under the influence of gravity. The pressure gradient is affected by the density difference between riverine and oceanic waters (Valle-Levinson, 2011). Zhang and Savenije (2016) reported that dispersion is proportional to the density gradient, when $0.51 < K < 0.66$. Therefore, the gravitational flow

produced by the density difference between Chalna and Harbaria (Fig. 6) advances towards the ocean (Harbaria) from the salt plug area (Chalna) during the dry season. As a result, the density-induced gravitational circulation facilitated the import of relatively light, sea water moving on the surface toward the salt plug area and the export of the relatively heavy, high-salinity water of the salt plug area flowing near the bottom toward the ocean (Fig. 6). The density-driven flow reverses direction with the depth at the salt plug area; thus, the salt plug creates a zone of

inverse gravitational circulation between it and the coastal ocean.

In addition, during the dry season, the spatial variation of *K* demonstrates a gradual increase in *K* landward from the salt plug area (from Chalna to Rupsha Bridge, Figs. 5-6), similar to a positive estuary. The *K* value of approximately 0.65 around Chalna indicates the control of density-driven positive gravitational circulation for up-

estuary salt transport. Zhang and Savenije (2016) find that dispersion is driven by the longitudinal salinity gradient if *K* ranges from 0.51 to 0.66. The vertical profiles of salinity clearly indicate that the longitudinal density gradient drives a net volume near-bottom inflow to the Rupsha Bridge from Chalna and a stronger surface outflow to Chalna from the Rupsha Bridge (Fig. 6). This circulation is induced by the volume of fresh water added to the PRE from upstream. Riverine waters, which are less dense than oceanic waters, are forced to flow

seaward (Valle-Levinson, 2011). Because the water that flows from the Shibsa River estuary to Chalna through the Chunkhuri Channel is denser than the water moving from the upstream of the PRE, the water level at the Ruphsha Bridge is slightly higher than the mean water level. The resultant hydrostatic pressure near the water surface at the Rupsha Bridge is directed towards Chalna. Thus, a strong counteraction between the density-driven



and discharge-induced gravitational flows occurs landward of the salt plug. During the dry season (due to the

negligible river discharge), the density-driven circulation was induced by the tide; consequently, salt water

intrusion extended as far as ~96 km upstream from Chalna (Shaha and Cho, 2016). As a result, all materials

introduced into the estuary by river-side industries can advance upstream with the salt water during the dry

season, potentially creating water quality problems (Samad et al., 2015; Shaha and Cho, 2016). The circulation

landward of the salt plug resembled that of a typical estuary during the dry season.


### 3.4. Relationship between river discharge and Van der Burgh's coefficient ($K$)

$K$ values were plotted against river discharge to examine the influence of freshwater discharge on the spatial

variation in $K$ (Fig. 7). The $K$ values were nearly constant for all levels of freshwater discharge near Harbaria

(SEG1~3). On the other hand, $K$ depended on the freshwater discharge upstream (SEG4~12), with the coefficient

of determination ($r^2$) ranging from 0.40 to 0.72. Although previous studies (Gisen, 2015; Savenije, 1993, 2005)

reported that $K$ is a time-independent parameter, this study reveals that $K$ may, in fact, be a time-dependent value

(Fig. 7), as suggested by Shaha and Cho (2011). Thus, gravitational circulation and tide-driven dispersive salt

flux differed with changing river discharge levels. Furthermore, the rates of change in the salt content for various

levels of river discharge in the PRE can be represented on the basis of the spatially varying value of $K$.

$K$ values calculated with Eq. (6) for different levels of river discharge did not lie within the feasible range of

$0<K<1$, as shown in Fig. 8. However, the spatially different $K$ values determined from Eq. (7) were within the

recommended range. Moreover, these values described the spatial variation of the salt transport mechanisms

reasonably well in the PRE during the dry and wet seasons. Salt transport was influenced by density-driven

mixing mechanisms in the central regimes of the large PRE, where salt plug occurred during the dry season.



The river discharge in the Schelde estuary is large compared to the tidal flow (Savenije, 2005). In the upper

reaches of the Schelde estuary, river discharge is largely responsible for the considerable tidal damping which

occurs. Therefore, density-driven mixing is prominent upstream from 60 km to 100 km in the Schelde Estuary

(Savenije, 2005). By contrast, tidal mixing mainly controls the salt transport landward, up to 60 km from the

mouth of the Schelde Estuary (Savenije, 2005). Therefore, a single value of $K$ (0.25) cannot represent the spatial

variation of both the tide-driven and density-driven mixing mechanisms in the Schelde Estuary (Savenije, 2005).

Therefore, one would expect a lower value of $K$ between 0.51 and 0.66 (Zhang and Savenije, 2016) for the salt

plug area as compared to that at the periphery ($K\sim0.74$) to describe the spatial variation of density-driven salt

transport mechanisms obtainable from Eq. (7). Thus, the $K$ values of Eq. (7) described the mixing processes well

at the salt plug area during the dry season. In addition, during the wet season, gravitational circulation was almost

entirely dominant over tidal dispersion in the central regimes of the PRE, thus efficiently lessening salt transport

upstream due to the high river discharge level. Therefore, it is clear that spatially-varying time-dependent $K$

values are indeed required to explain the nature of the spatially varying salt transport mechanisms in a salt plug

estuary with a varying geometry.

## 4. Conclusion

We determined the spatially varying Van der Burgh's coefficient along the axis of the PRE using high-resolution

salinity data to characterize salt flux mechanisms in the dry and wet seasons. In the wet season, salt transport was

dominated downstream (landward up to 10 km from Harbaria) by both tide-driven and gravitational circulation

during spring and neap tides. Upstream (over 10 km from Harbaria), gravitational circulation was almost entirely

dominant over tidal dispersion, effectively diminishing salt transport upstream during spring and neap tides due



to the high river discharge level ( >750 m$^3$s$^{-1}$). On the other hand, during the dry season, when the salt plug

formed due to the decreasing river discharge upstream, $K$ values were reduced to those of the salt plug area

(~0.65) from the periphery (~0.74), describing the density-driven salt transport mechanisms reasonably well at

the salt plug area during the spring and neap tides. Density-induced inverse gravitational circulation between the

salt plug and the coastal ocean caused inverse estuarine circulation, relatively high-salinity bottom water flows

towards the coastal ocean from the salt plug area and relatively low-salinity surface water flows to the salt plug

area from the ocean. In contrast, positive gravitational circulation between the salt plug and the river area drove

high-salinity bottom water upstream. Thus, this result shows that $K$ also works in the opposite direction of the salt

plug area, where gravitational circulation is reversed. In addition, our results demonstrated that $K$ not only varied

spatially but is also dependent on the river discharge level.


Density-driven circulation in the salt plug area was induced by the tide during the dry season due to the negligible

river discharge, causing salt water intrusion of ~96 km upstream. This indicates that salt water can also carry

materials upstream that were introduced into the estuary by industries situated along the river. Moreover, if

pollutants are introduced upstream, they may reside in the estuary until the next wet season, much to the

detriment of the Pasur River estuarine ecosystem. Thus, our understanding of salt transport mechanisms may

have far reaching implications and may contribute to a better understanding of the spatial and temporal

distributions of pollutants, nutrients and biota within large tropical estuaries.




**Acknowledgement**

This work was supported by the International Foundation for Science (IFS), Sweden (W/5414-1). Y.-K. Cho partly

supported by the project entitled "Long-term change of structure and function in marine ecosystems of Korea,"

which was funded by the Ministry of Oceans and Fisheries, Korea. The first author would like to thank Dr. Md.

Shahidul Islam, Universiti Malaysia Terengganu in Malaysia for his inspiring advice to write a research proposal

for an IFS grant. The authors would like to thank N. Imtiazy and K. Pramanik for their support during the data

collection process. We sincerely acknowledge and thank the Mongla Port Authority and all of our field assistants

for their constant support during the field work.

Table 1. Sampling scheme:

| Seasons | Tide | Longitudinal CTD transects | River discharge ($m^3s^{-1}$) |
|---------|------|----------------------------|-------------------------------|
| Dry | Spring | 26 December, 29 March, 29 April, and 13 June 2014 | 28.7 |
|  | Neap | 24 February and 09 May 2014 | 9.2 |
| Wet | Spring | 12 July and 24 October 2014 | 803.2 |
|  | Neap | 22 August 2014 | 1606.5 |




**Fig. 1.** (**a**) Map of the complex topographical features of the multi-channel Pasur River-Shibsa River estuarine system in the southwestern coastal zone of Bangladesh. (**b**) Conductivity-temperature-depth (CTD) recorder stations are shown as red solid circles (●) in the Pasur River. The Xs (✖) denote the locations of the tidal stations at Hiron Point and Mongla Port. (**c**) The export of salt water from the Shibsa River Estuary to the Pasur River Estuary through the Chunkhuri Channel, creating a salt plug. The map was generated using Golden Software Surfer 9.0 (www.goldensoftware.com).






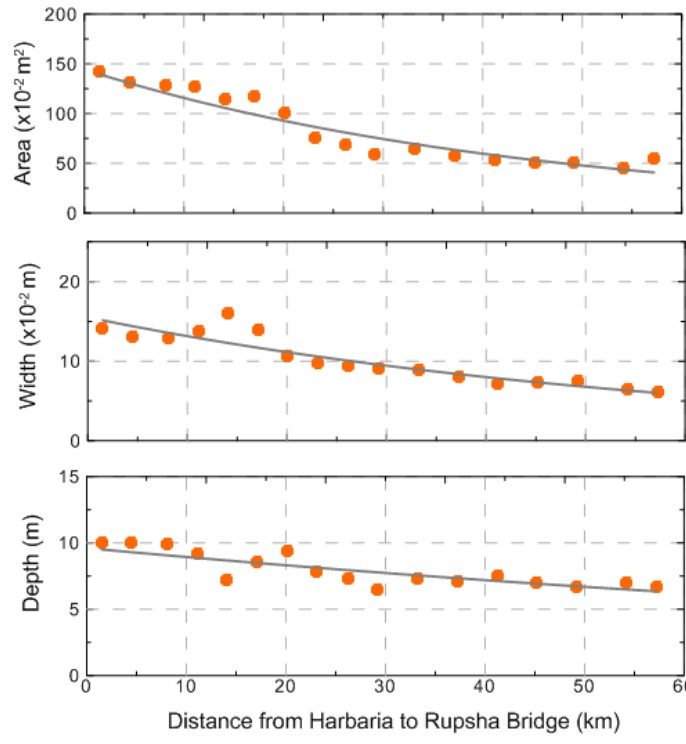

**Fig. 2.** Cross-sectional area, width and depth of all CTD stations in the Pasur River Estuary.







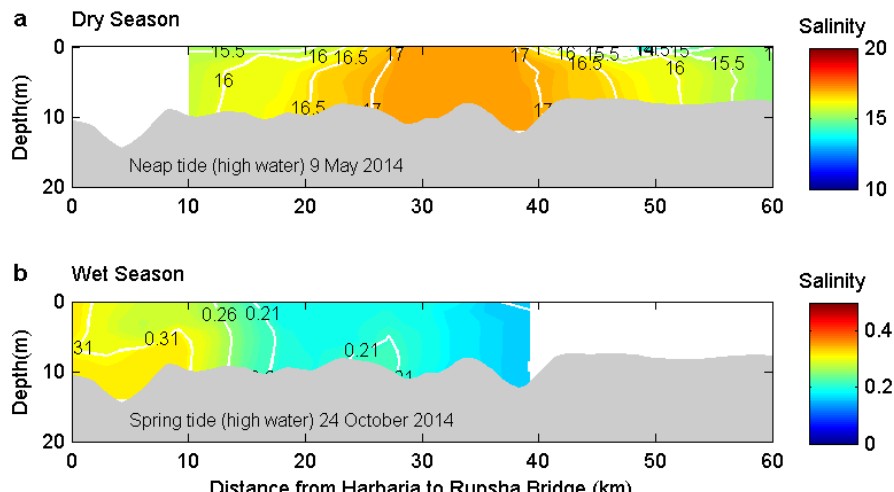


**Fig. 3.** (**a**) Vertical salinity sections obtained along the main axis of the Pasur River Estuary during the dry season. A salt plug developed near Chalna, 34 km upstream of Harbaria. (**b**) Vertical salinity sections obtained along the main axis of the Pasur River Estuary during the wet season. The salt plug disappeared and a typical estuarine system

developed.





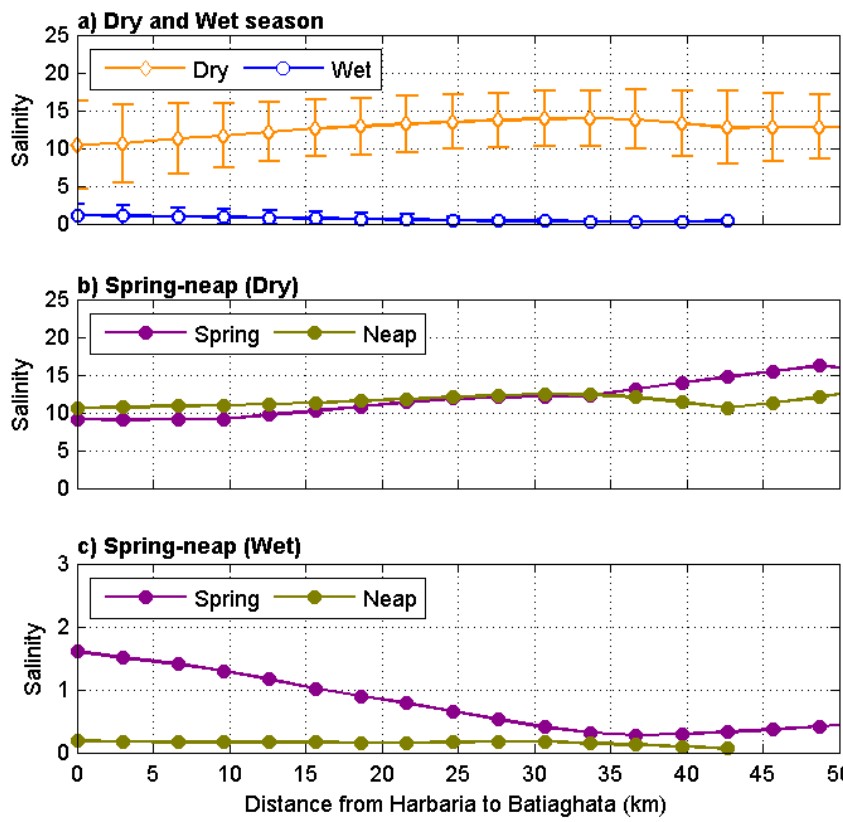

**Fig. 4.** Depth-averaged salinity distribution at high water during neap and spring tides in the Pasur River Estuary

in the wet and dry seasons.







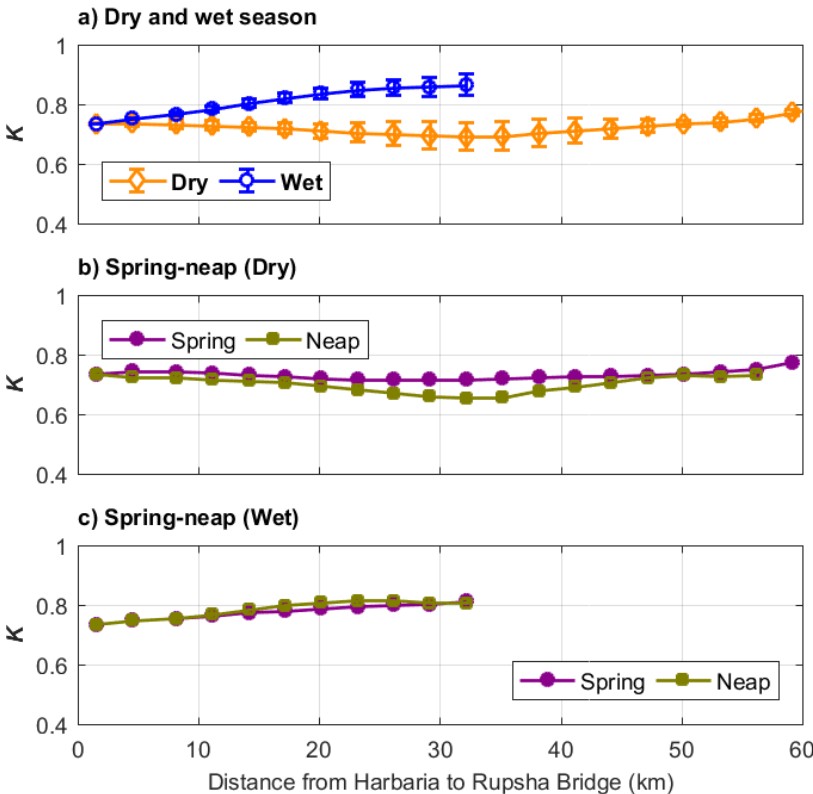

**Fig. 5.** Spatial variation of Van der Burgh's coefficient ($K$) along the Pasur River Estuary. If $K < 0.3$, up-estuary salt transport is entirely dominated by tide-driven mixing. If $K > 0.8$, up-estuary salt transport is almost entirely dominated by gravitational circulation. If $0.3 < K < 0.8$, both gravitational circulation and tide-driven circulation

contribute to the up-estuary transport of salt.






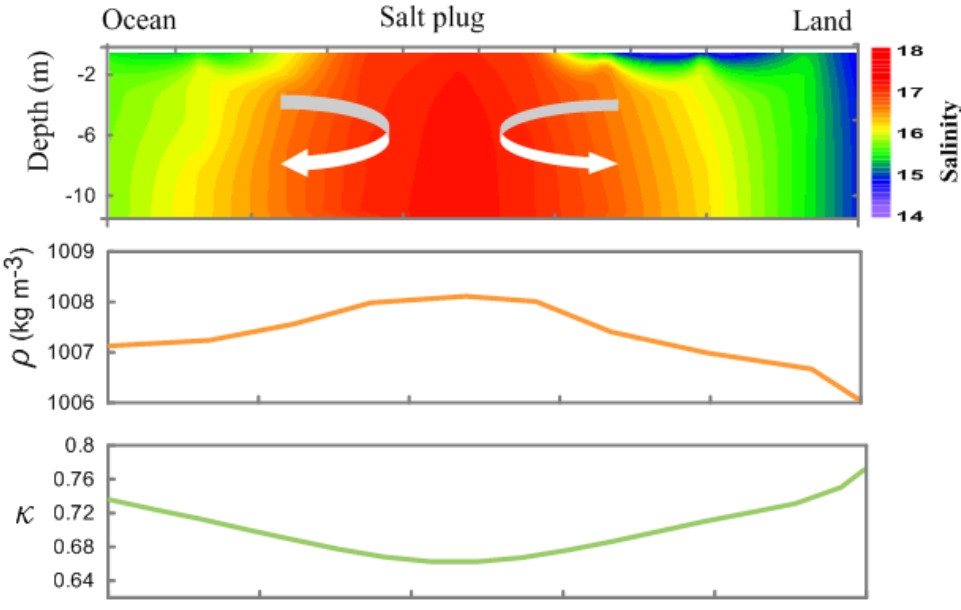

**Figure 6.** Conceptual diagram of an idealized baroclinic flow in a salt plug. During the dry season when a salt
plug is formed, a longitudinal density gradient produces a zone of inverse gravitational circulation between the
salt plug and the coastal ocean, and a zone of positive gravitational circulation near the river area. Van der
Burgh's coefficient ($K$) indicates a gradual increase seaward and landward from the center of the salt plug,
similar to inverse and positive estuaries, respectively. This shows that $K$ works in the opposite direction, when
gravitational circulation is reversed.




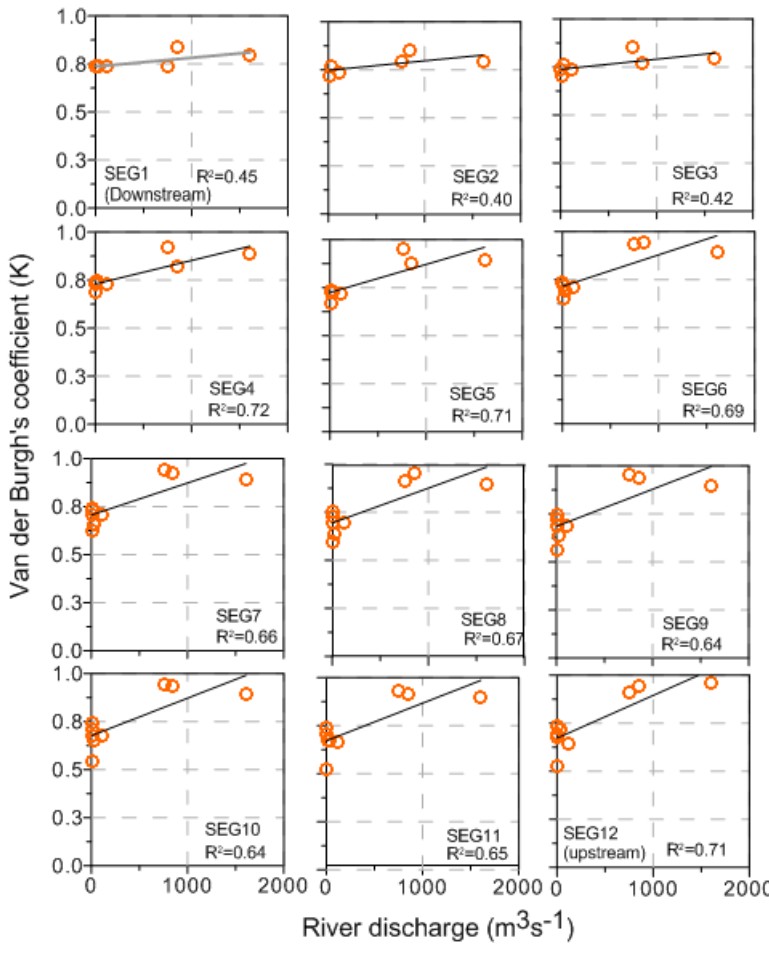

**Fig. 7.** Plots of Van der Burgh's coefficient (*K*) against river discharge for different segments of the Pasur River

estuary.







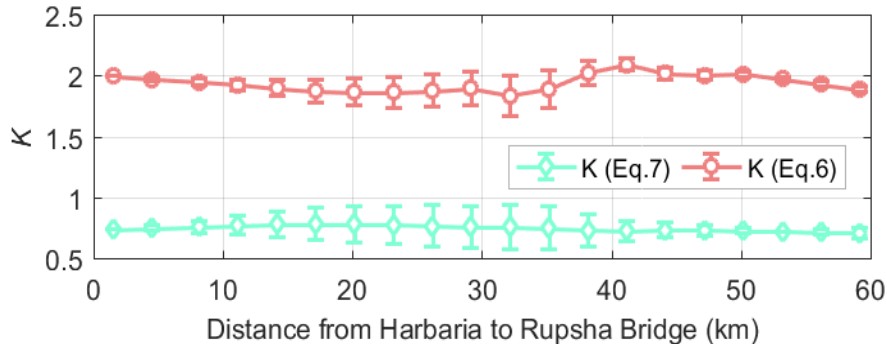

**Fig. 8.** Spatial variation of Van der Burgh's coefficient ($K$) as calculated by Eqs. (6) and (8).

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
