# Peer review of "Spatiotemporal Variation of Van der Burgh's coefficient in a salt plug estuary"

_Hydrology and Earth System Sciences, 2017_

## Referee Comment (RC1) · Z. Zhang (Referee) · 15 Apr 2017

The authors demonstrated that the empirical Van der Burgh coefficient is spatiotemporally varying. By using a modified equation, the authors calculated this K along the estuary. Additionally, different situations during spring and neap tide in the dry and wet season were considered. The issue in a salt plug estuary is interesting. And it is indeed very interesting to test whether K functions in an inverse salinity gradient. However, the manuscript has several major issues.

1. To distinguish between density- and tidal-driven dispersion, the authors used the empirical Van der Burgh coefficient. Their trick is to use an exponential transformation in K ∈ (0, 1). However, the relation between a certain K range and the dispersion mechanism is vague. For instance, K~0.8 is considered gravitational circulation, and during the dry season, K ∈ (0.65, 0.74) is also considered density-driven. While K ∈ (0.7, 0.8) is considered both density and tidal driven matter (e.g., Line 253, Page 12).

And why do you consider that gravitational circulation in the upper part of the estuary corresponds with weak mixing processes? (Line 214-215, Page 10). Could you please

discuss this more?

The authors mentioned that the 'salt transport mechanism varies' or 'the K values describe the spatial variation of the salt transport mechanisms well' (e.g., Line 109, Page 5. Line 313, 319, Page 14. Line331, Page 15. Line 347, Page 16) which are unconvincing. Basically, the K varies (slightly) along the estuary and in time, but the mechanism is almost entirely density-driven gravitational circulation (besides from Harbaria 10 km upstream in wet seasons). Could you please discuss this more?

2. Line 253-259, Page 12. From Harbaria to 10 km upstream, K ranged from 0.7 to 0.8, both tide and density drive the mixing during the wet season. And you made another conclusion that gravitational circulation is dominant in the next sentence, which is not consistent. Moreover, you explained the tide effect by introducing tidal amplification, which happened from Hiron Point to Mongala Port (34 km from Harbaria where the salinity intrusion limits).

3. The conclusion the authors made about wet/dry season and spring/neap tide effects is not strong. The number of

events is small. Moreover, the author just compared dry/wet periods and spring/neap tides separately. Whereas in reality, those two parameters define the stratification together. Also the discharge varies a lot between the dry and wet season while the difference between neap and spring tide is small. The effect of neap/spring variation may be affected by the discharge even during the same season.

4. In the manuscript, the authors used words like 'density-induced gravitational circulation induced by the tide', 'discharge-induced', 'tidal-induced density-driven circulation'. Density-driven or tide-driven, or something else? It is really confusing. Density differences (stratification) result from the balance between river discharge and tide. It is the Richardson number that determines it (the ratio of potential energy of buoyant fresh water to kinetic energy of the tide). In well-mixed estuaries tide-driven dispersion is dominant. In more stratified estuaries density-driven dispersion is dominant.

5. Line 234-238, Page 11. Did you use an error-bar for describing the depth-averaged salinity range? And what

causes the error in Figure (4a)? You mentioned that during neap tide in the wet season the gravitational circulation is enhanced, but from the figure (4c), the water is almost fresh from Harbaria to upstream. How does the gravitational circulation happen?

Minor comments:

1 The modified equation to account for the exponential variation in estuarine widths, especially in a small, narrow estuary (e.g., Line 76, Page 4). But in narrow estuaries, the exponential varying of width is not strong. Could you please discuss this more?

2 Line 80, Page 4. What do you mean by mentioning '...a time-independent factor...and geometries'?

3 Line 184-195, Page 8-9. The tide-driven dispersion is $D_t \partial S / \partial x$ (Savenije, 2005) instead of $D \partial S / \partial x$. And why S ($=S_0$) is constant in equation (4)? In addition, could you please derive (5) in detail?

4 Line 200, Page 9. If the PRE is partially mixed, is the equation in Line 199 still working?

5 Line 217-218, Page 10. The calculating equations are

different, so there is no need to mention the range with other results. Also Line 278 and 290, Page 13. Line 329, Page 15.

6 Line 259-261 and 272, Page 12. The difference between spring and neap tide in the wet season is smaller than that in the dry season. But the author stressed the former one and mentioned that the latter one is not significant. Could you please discuss this more?

7 Line 311, Page 14. '$r^2$' should be '$R^2$'.

---

## Referee Comment (RC2) · Z. Wang (Referee) · 18 Apr 2017

The manuscript presents an interesting study showing that the Van der Burgh's coefficient is spatially varying in a salt plug estuary. It is also shown that the coefficient is different in dry and wet season. As stated in the manuscript, this is the first study to Van der Burgh's coefficient in a salt plug estuary. Therefore the manuscript can in potential become an interesting paper. However, there is a serious problem with the method used in the study. The method presented is based on the salt intrusion theory for a "normal" estuary and missed an essential physical process in a salt plug estuary, i.e. evaporation. Without evaporation the inverse salinity gradient in a salt plug estuary cannot be generated. Evaporation has the effect that the residual discharge Q is no more constant along the estuary and will even turn it to be landwards directed in the part with inverse salinity gradient. In the manuscript Q is presented as constant along the estuary. This means that the results from the analysis cannot be used, especially for dry season. I would recommend the authors carrying out the analysis again after determining the spatially varying Q (residual discharge) by taking into account evaporation and precipitation.

More in detail, I cannot fully follow the presented theory after Eq.(3). Of course, this is partly due to the fact that I do not have the complete overview of the related literature. I have the following problems:

1. On line 84 just below Eq.(3), the expressions between brackets are fluxes, and not dispersion coefficients as suggested.
2. Why is $S_0$ instead of $S(x)$ used in Eq.(4)?
3. What is the motivation from Eq.(6) to Eq.(7)? The authors refer to their paper in (2011), but I could not find the motivation in that paper either. That there seems to be a paradox with the relation between K and $\upsilon$ (resulting is K>1) was already pointed out by Savenije (2005). Using the exponential function of $\upsilon$ indeed solves the paradox, but what is the rationale behind this solution?

I wonder why the authors do not just determine K directly by first determining D(x) from S(x).

---

## Author Comment (AC1) · 4 Jun 2017

Dear Anonymous Referee#1,

We are very much grateful for your valuable and fruitful comments to improve our manuscript (hess-2017-76). The referee comment is given in blue font and the answer in black font.

**1.1.** To distinguish between density- and tidal-driven dispersion, the authors used the empirical Van der Burgh coefficient. Their trick is to use an exponential transformation in K ∈ (0, 1). However, the relation between a certain K range and the dispersion mechanism is vague. For instance, K~0.8 is considered gravitational circulation, and during the dry season, K ∈ (0.65, 0.74) is also considered density-driven. While K ∈ (0.7, 0.8) is considered both density and tidal driven matter (e.g., Line 253, Page 12).

Answer: Van der Burgh's coefficient K is one parameter used to describe the nature of salt transport mechanisms (both tide driven and density-driven dispersion) in estuaries (Savenije, 2006). This coefficient determines the relative weight of these mechanisms (Savenije, 2006). If K is small, then tide-driven mixing is dominant in transporting salt. If K approaches 1, gravitational circulation is dominant in transporting salt. Tide driven dispersion dominates near the mouth of the Pungue and Maputo Estuaries (Savenije, 2005). In contrast, density driven mixing dominates upstream from the location of the strong salinity gradient. The value of K obtained for both estuaries is 0.3, which implies only the tide-driven dispersion mechanism transported salt in the Pungue and Maputo Estuaries. In reality, two mechanisms exist along the estuaries, indicating that no single value of K can describe the nature of salt transport in estuaries, but it would vary along an estuary. Considering this limitation of K, the $K$ value has been scaled on the basis of the $v$ value, and it ranges from 0 to 1 (Shaha and Cho, 2011). If $K$ <0.3, the total salt transport is driven by diffusive processes (e.g., tidal mixing), as in unidirectional net flows. If $K$ ~> 0.8, up-estuary salt transport is controlled by advection, i.e., by discharge-driven gravitational circulation. If $0.51 < K < 0.66$, the dispersion is proportional to the salinity gradient, meaning it is driven by the longitudinal density gradient (Zhang and Savenije, 2016).

Revised text for the revised manuscript is as follows:
Discharge-driven gravitational circulation greatly weakened salt transport due to high discharge levels ($>750$ m$^3$s$^{-1}$) in the wet season when $K > 0.8$.

**1.2.** Why do you consider that gravitational circulation in the upper part of the estuary corresponds with weak mixing processes? (Line 214-215, Page 10). Could you please discuss this more?

Answer:

"In this case, mixing processes are weak, as in a highly stratified estuary (Valle-Levinson, 2010)".

This is not the result of this study. In the methodology, we have tried to explain that mixing process is weak when an estuary is highly stratified. I guess the phrase ("in this case") makes this confusion. We will delete this in the revised version.

**1.3.** The authors mentioned that the 'salt transport mechanism varies' or 'the K values describe the spatial variation of the salt transport mechanisms well' (e.g., Line 109, Page 5. Line 313, 319, Page 14. Line331, Page 15. Line 347, Page 16) which are unconvincing. Basically, the K varies (slightly) along the estuary and in time, but the mechanism is almost entirely density-driven gravitational circulation (besides from Harbaria 10 km upstream in wet seasons). Could you please discuss this more?

Answer: We revised this as follows:

**Line 313, 319, Page 14. (Relation between river discharge and K)**

Although previous studies (Gisen, 2015; Savenije, 1993, 2005) reported that $K$ is a time-independent parameter, this study reveals that $K$ is not only a time-dependent value (Fig. 7), but also clearly shows an inverse and positive gravitational circulation from the salt plug, respectively (Fig. 6). Thus, discharge-driven and density-driven gravitational salt flux differed with changing river discharge levels.

$K$ values calculated with Eq. (6) for different levels of river discharge did not lie within the feasible range of $0<K<1$, as shown in Fig. 8. However, the spatially different $K$ values determined from Eq. (7) were within the recommended range. Moreover, these values described the spatial variation of the salt transport mechanisms in the PRE during the dry and wet seasons. Salt transport was influenced by density-driven mixing mechanisms in the central regimes of the large PRE, where salt plug occurred during the dry season. This density-driven mechanism clearly showed an inverse and a positive gravitational circulation seaward and landward from the salt plug area, respectively.

**Line331, Page 15. (Relation between river discharge and K)**

A single value of $K$ (0.25) cannot represent the spatial variation of both the tide-driven and density-driven mixing mechanisms in the Schelde Estuary (Savenije, 2005). Therefore, one would expect a lower value of $K$ between 0.51 and 0.66 (Zhang and Savenije, 2016) for the salt plug area to describe the density-driven salt transport mechanisms obtainable from Eq. (7). Thus, the $K$ values of Eq. (7) described the density-driven salt transport mechanisms at the salt plug area during the dry season.

**Line 347, Page 16 (Conclusion)**

In the wet season, discharge-driven gravitational circulation was almost entirely dominant over tidal dispersion, effectively diminishing salt transport upstream during spring and neap tides due to the high river discharge level ($>750$ m$^3$s$^{-1}$). On the other hand, during the dry season, when the salt plug formed due to the decreasing river discharge upstream, $K$ values were reduced to those of the salt plug area (~0.65) from the periphery (~0.74), describing the density-driven salt transport mechanism at the salt plug area with negative and positive estuarine circulation seaward and landward from salt plug area, respectively, during the spring and neap tides. Inverse gravitational circulation between the salt plug and the coastal ocean caused outflows of high-salinity bottom water towards the coastal ocean from the salt plug area and inflows of relatively low-salinity surface water to the salt plug area from the ocean.

2. Line 253-259, Page 12. From Harbaria to 10 km upstream, K ranged from 0.7 to 0.8, both tide and density drive the mixing during the wet season. And you made another conclusion that gravitational circulation is dominant in the next sentence, which is not consistent. Moreover, you explained the tide effect by introducing tidal amplification, which happened from Hiron Point to Mongala Port (34 km from Harbaria where the salinity intrusion limits).

**Answer: We revised this as follows:**

Discharge-driven gravitational circulation greatly weakened salt transport due to high discharge levels ($>750$ m$^3$s$^{-1}$) in the wet season when $K > 0.8$.

3. The conclusion the authors made about wet/dry season and spring/neap tide effects is not strong. The number of events is small. Moreover, the author just compared dry/wet periods and spring/neap tides separately. Whereas in reality, those two parameters define the stratification together. Also the discharge varies a lot between the dry and wet season while the difference between neap and spring tide is small. The effect of neap/spring variation may be affected by the discharge even during the same season.

Answer: In the same season, spring-neap variation was not significant when river discharge was not varied significantly (Fig. 4b). However, spring-neap variation can be affected by different river discharge in the same season (Fig. 4c).

4. In the manuscript, the authors used words like 'density-induced gravitational circulation induced by the tide', 'discharge-induced', 'tidal-induced density-driven circulation'. Density-driven or tide-driven, or something else? It is really confusing. Density differences (stratification) result from the balance between river discharge and tide. It is the Richardson number that determines it (the ratio of potential energy of buoyant fresh water to kinetic energy of the tide). In well-mixed estuaries tide-driven dispersion is dominant. In more stratified estuaries density-driven dispersion is dominant.

Answer:

Estuarine circulation represents the interaction among the contributions from gravitational circulation, tidal residual circulation, and circulation driven by tidally asymmetric vertical mixing. In turn, gravitational circulation is driven by river discharge and density gradients (Valle-Levinson, 2011). Gravitational circulation tends to be dominant in many estuaries and can be classified according to the basin's morphology or origin, to its water balance, or to the competition between tidal forcing and river discharge(Valle-Levinson, 2011). Therefore, we used the terms density-driven and discharge-driven gravitational circulation.

In the revised version, we resolved these problems by using tide-driven, discharge-driven, and density-driven terms instead of tide-induced, density-induced and discharge-induced terms.

5. Line 234-238, Page 11. Did you use an error-bar for describing the depth-averaged salinity range? And what causes the error in Figure (4a)? You mentioned that during neap tide in the wet season the gravitational circulation is enhanced, but from the figure (4c), the water is almost fresh from Harbaria to upstream. How does the gravitational circulation happen?

Answer: Thank you very much for this constructive and valuable suggestion. In the revised version, we described the depth-average salinity range considering the error-bar. In addition, we corrected the inconsistent explanation of gravitational circulation between text and figure (4c). The revised text is as follows:

The depth-averaged salinity ranged between 6 and 17 in the dry season. The vertical salinity sections obtained along the main axis of the PRE during the dry season (December, February, March, April, May and June) in 2014. Minimum salinity of 6 was found in February whereas maximum salinity of 17 was found in June (Shaha and Cho, 2016). In addition, a salt plug developed near Chalna (34 km upstream of Harbaria) (Shaha and Cho, 2016). This salt plug started to develop in transit during the dry winter season (December and February). The relative water level variation between the SRE and the PRE during the dry season exerted hydrostatic pressure towards the PRE from the SRE and facilitated an export of salt water from the SRE to the PRE through the Chunkhuri Channel and thus created this salt plug. This salt plug persists for several months (December-June). Therefore, the error bar was higher during the dry season than the wet season. The salt plug disappeared in the wet season, and developed a typical estuarine system. As a result, the error-bar becomes small during the wet season (Fig. 4a). The depth-averaged salinity did not vary significantly between spring and neap tides in the dry season (Figs. 4a-b). In contrast, during the wet season, the salinity varied between 0.15 and 3.0 (Figs. 4a and c). The salinity was lower during neap tide than during spring tide in the wet season, most likely due to higher river discharge levels.

**Minor comments:**
1. The modified equation to account for the exponential variation in estuarine widths, especially in a small, narrow estuary (e.g., Line 76, Page 4). But in narrow estuaries, the exponential varying of width is not strong. Could you please discuss this more?
**Answer:**
Shaha and Cho (2011), who suggested a modified equation to account for the exponential variation in estuarine widths, examined the spatial variability of $K$ along the axis of a small, narrow estuary with a large salinity gradient of 1.4 psu km$^{-1}$. In narrow Sumjin estuary, both the large spatial salinity gradient and exponentially varying width are responsible for spatial variation of K and salinity distribution (Shaha and Cho, 2011).

2. Line 80, Page 4. What do you mean by mentioning '…a time-independent factor…and geometries'?
**Answer:**
This is the findings of Gisen (2015). I guess, due to absence of reference here, it makes confusion. In the revised version, we added the reference as follows to avoid this confusion.
Revised text is as follows:
Nonetheless, debate continues regarding the use of $K$ for an estuary, i.e., whether this value should be constant (Savenije, 2005) or spatially varying (Shaha and Cho, 2011) and/or whether it can serve as a time-independent factor for varying river discharges (Gisen, 2015) and depend on geometries (Gisen, 2015).

3. Line 184-195, Page 8-9. The tide-driven dispersion is $Dt\, \partial S/\partial x$ (Savenije, 2005) instead of $D\partial S/\partial x$. And why S (=S0) is constant in equation (4)? In addition, could you please derive (5) in detail?

Answer:

Eq.(4) will be corrected as Shaha and Cho (2011). Please see the paper (Shaha and Cho, 2011).

4. Line 200, Page 9. If the PRE is partially mixed, is the equation in Line 199 still working?

Answer:

Well-mixed ($n_s < 0.1$) conditions were observed from Harbaria to Batiaghata during the dry season, with slightly partially mixed conditions near the confluence between the Batiaghata Channel and the PRE due to the advection of freshwater from Batiaghata channel (Fig. 1a). By contrast, during the wet season, well-mixed ($n_s < 0.1$) conditions were observed from Harbaria to Mongla Port and slightly partially mixed conditions upstream from Mongla Port. In addition, the estuary showed well-mixed conditions upstream from Harbaria during spring and neap tides except for slightly partially mixed conditions near the confluence between the Batiaghata Channel and the PRE during neap tides in the dry season (Fig. 1b). By contrast, during the wet season, slightly partially mixed conditions were observed along the PRE during neap tides and well-mixed conditions during spring tides (Fig. 1c). Freshwater discharged from Batiaghata Channel into the PRE may be responsible for this slightly partially mixed condition which can be considered as negligible.

[Figure]

**Fig. 1.** Spatial variation of the stratification parameter ($n_s$) at high water during spring and neap tides in the dry and wet seasons along the Pasur River Estuary.

5. Line 217-218, Page 10. The calculating equations are different, so there is no need to mention the range with other results. Also Line 278 and 290, Page 13. Line 329, Page 15.

Answer: We used this reference to represent the density-driven circulation considering the K values of Gisen (2015) which coincide with this study.

6. Line 259-261 and 272, Page 12. The difference between spring and neap tide in the wet season is smaller than that in the dry season. But the author stressed the former one and mentioned that the latter one is not significant. Could you please discuss this more?

Answer: The spatial variation of $K$ between spring and neap tide in the wet season is smaller than that in the dry season (Fig. 4c). We agree with this valuable comments of the reviewer and we removed this explanation to avoid inconsistency.

7. Line 311, Page 14. 'r2' should be 'R2'.

Answer: we have fixed in the final version.

**References**

Gisen, J. I. A.: Prediction in ungauged estuaries, TU Delft, Delft University of Technology, 2015.

Savenije, H. H.: Comment on "A note on salt intrusion in funnel-shaped estuaries: Application to the Incomati estuary, Mozambique" by, Estuarine, Coastal and Shelf Science, 68, 703-706, 2006.

Savenije, H. H.: Predictive model for salt intrusion in estuaries, Journal of Hydrology, 148, 203-218, 1993.

Savenije, H. H.: Salinity and tides in alluvial estuaries, Elsevier, 2005.

Shaha, D. and Cho, Y.-K.: Determination of spatially varying Van der Burgh's coefficient from estuarine parameter to describe salt transport in an estuary, Hydrology and Earth System Sciences, 15, 1369-1377, 2011.

Shaha, D. C. and Cho, Y. K.: Salt Plug Formation Caused by Decreased River Discharge in a Multi-channel Estuary, Scientific Reports, 6, 2016.

Valle-Levinson, A.: Classification of estuarine circulation, Treatise on estuarine and coastal science, 1, 75-86, 2011.

Valle-Levinson, A.: Contemporary issues in estuarine physics, Cambridge University Press, 2010.

Zhang, Z. and Savenije, H. H.: The physics behind Van der Burgh's empirical equation, providing a new predictive equation for salinity intrusion in estuaries, Hydrology and Earth System Sciences Discussions, 2016. 2016.

---

## Author Comment (AC2) · 4 Jun 2017

Dear Anonymous Referee#2,

We are very much grateful for your valuable and fruitful comments to improve our manuscript (hess-2017-76). The referee comment is given in blue font and the answer in black font.

1.The method presented is based on the salt intrusion theory for a "normal" estuary and missed an essential physical process in a salt plug estuary, i.e. evaporation. Without evaporation the inverse salinity gradient in a salt plug estuary cannot be generated. Evaporation has the effect that the residual discharge Q is no more constant along the estuary and will even turn it to be landwards directed in the part with inverse salinity gradient. In the manuscript Q is presented as constant along the estuary. This means that the results from the analysis cannot be used, especially for dry season. I would recommend the authors carrying out the analysis again after determining the spatially varying Q (residual discharge) by taking into account evaporation and precipitation.

Answer:

The salt pug results from strong evaporation in tropical estuary. However, the salt plug in our study area does not result from the evaporation but from the intrusion of saline water from a tributary (Shaha and Cho, 2016).

The changes of salinity by evaporation and precipitation are negligible in our estuary. The evaporation rate ($e$) in the salt plug area (A ~ 40 km$^2$) of the Pasur river estuary (Shaha et al., 2017; Shaha and Cho, 2016) is ranged between 0.02 and 0.04 cm day$^{-1}$ in the dry season (Fig.1) with a minimum value of about 0.025 cm day$^{-1}$ in wet season (August). This yields an estimated loss of water by evaporation ($E=eA$) (Ribbe, 2006; Wolanski, 1986). The river discharge is greater than the evaporation loss in the salinity maximum zone (Fig. 2 and 3).

[Figure]

Fig. 1. Monthly average evaporation rate ($e$) from the year 2013 to 2014.

[Figure]

Fig. 2. Monthly average Evaporation ($E$) from the year 2013 to 2014.

[Figure]

Fig. 3. Monthly average Evaporation ($E$) for the dry season. The volume of river discharge is larger by about 300 times than the evaporation during the dry season.

2. On line 84 just below Eq. (3), the expressions between brackets are fluxes, and not dispersion coefficients as suggested.

Answer:

$D_t$ is horizontal diffusive (=tide driven) dispersion (Savenije, 2005, Eq. 4.31, Page no. 134).

We will correct it in the revised version.

3. Why is $S_0$ instead of $S(x)$ used in Eq. (4)?

Answer:

This is a typo. This will be corrected in the revised version following Shaha and Cho (2011).

4. What is the motivation from Eq. (6) to Eq. (7)? The authors refer to their paper in (2011), but I could not find the motivation in that paper either. That there seems to be a paradox with the relation between K and (resulting is K>1) was already pointed out by Savenije (2005). Using the exponential function of indeed solves the paradox, but what is the rationale behind this solution?

Answer:

Savenije (2005) showed an equation $K=1/v$ (page 135, below eq. 4.33). We first calculated $v$ (x) from Eq. (4) and then calculated K from Eq. (6). Prior to this calculation, $K$ was calculating using a predictive equation 5.71 (page 169, Savenije 2005) which is constant value along the estuary. In addition, the K values calculated from Eq (6) does not lie between 0 and 1. If we use an exponential function with the proportion of tidal dispersion to total dispersion following the theory of McCarthy (1993), we can limit the range between 0 and 1 as well as the values express the relative contribution of salt transport mechanism (Savenije, 2006) along an estuary. For example, (Shaha and Cho, 2011) found a spatial variation of different salt transport mechanisms in the Sumjin estuary where the salinity varied from 31 psu at mouth to 1 psu at 25 km upstream from mouth with a salinity gradient of about 1.4 psu $km^{-1}$.

5. I wonder why the authors do not just determine K directly by first determining D(x) from S(x).

We first determined D(x) to calculate $v$(x) using Eq. (4). As the equation D(x) has shown after Eq.(6) and it makes confusion to reader, we will revise it in the final version and put it after Eq. (4) to avoid this confusion.

**References**

Ribbe, J.: A study into the export of saline water from Hervey Bay, Australia, Estuarine, Coastal and Shelf Science, 66, 550-558, 2006.

Savenije, H. H.: Comment on "A note on salt intrusion in funnel-shaped estuaries: Application to the Incomati estuary, Mozambique" by, Estuarine, Coastal and Shelf Science, 68, 703-706, 2006.

Shaha, D. and Cho, Y.-K.: Determination of spatially varying Van der Burgh's coefficient from estuarine parameter to describe salt transport in an estuary, Hydrology and Earth System Sciences, 15, 1369-1377, 2011.

Shaha, D. C., Cho, Y.-K., Kim, B. G., Sony, M. R. A., Md, S. R. K., and Islam, F.: Spatiotemporal Variation of Van der Burgh's coefficient in a salt plug estuary, 2017. 2017.

Shaha, D. C. and Cho, Y. K.: Salt Plug Formation Caused by Decreased River Discharge in a Multi-channel Estuary, Scientific Reports, 6, 2016.

Wolanski, E.: An evaporation-driven salinity maximum zone in Australian tropical estuaries, Estuarine, Coastal and Shelf Science, 22, 415-424, 1986.

---

## Editor Comment (EC1) · HHG Savenije (Editor) · 8 Jun 2017

I have appreciated your detailed replies to the reviewers. Indeed, referee#2 was not aware that the salt plug originated from a connecting side-channel. Maybe it is a good idea to stress more explicitly that this is a very special case of a "quasi-hypersaline" estuary where the salt plug is not caused by evaporation, but rather by the lateral input of more saline water from a side channel. Your reply to the comment is fine. Please make sure that you properly address all the other comments in your final submission.

---

## Referee Report (RR1)

The response from the authors to my review resolved my major concern about the manuscript. After reading the response from the authors to the reviews and the revised manuscript I still have the following questions concerning the parameters D, $D_i$ and $D_t$:

1. What is the relation between $D_i$ (only used in Eq.1) and D and/or $D_t$?
2. What is D, a diffusion coefficient with the dimension $L^2T^{-1}$ as suggested by Eq.(2) and line 200, or a flux as suggested in line 194 (salinity X velocity)?

I hope that the authors can resolve these issues before the manuscript is published.

---

## Author Response (AR2)

**Dear Anonymous Referee#1,**

We are very much grateful for your valuable and fruitful comments to improve our manuscript. The referee comment is given in blue font and the answer in black font.

1. The largest question in this manuscript is the paradox between the well-mixed estuary and salinity-driven mechanism. If the estuary is well-mixed, the density driven mixing is not strong. The studied estuary has low salinity gradient (Line 170, Page 7) and is well-mixed during spring and neap tides in the dry and wet seasons (Line201, Page 8). While in your description, gravitational circulation is dominant in both wet (e.g., Line 268-269, Page 11) and dry (e.g., Line 284-285, Line 299, Page 12; Line 358, Page 15) seasons. This may need some correction.

   Answer:
   To avoid confusion, we revised the sentence (line 169-170, page 7) as following.
   "The nominal distance between stations was approximately 3 km along the estuary".

   To avoid confusion, we delete the sentence (line 202-203, page 8).

   To avoid the confusion regarding gravitational circulation, we have already clarified about gravitational circulation in section 3.2 in the previous revised version (line 260-262, page 11). Gravitational circulation is driven by river discharge and density gradients (Valle-Levinson, 2011). Hereafter, we will use the terms density-driven and discharge-driven gravitational circulation.

   In addition, we added following sentence (line 233-235, page 10).
   "This salt plug, a region of maximum salinity, separates a zone of positive gravitational circulation near the river/estuary area and a zone of inverse gravitational circulation between the salt plug and the coastal ocean (Valle-Levinson, 2010)".

   We keep those (Line 268-269, Page 11; Line 284-285, Line 299, Page 12; Line 358, Page 15) as it is.

2. It is misleading in Line 248-251, Page 10. Actually, the depth-averaged salinity varies between spring and neap tides in dry season (up to ~4 psu), instead, the variation is less than ~1.5 psu in wet season.

Answer: We have revised as following (line 251-253, page 10):

The depth-averaged salinity varied upto ~ 4 psu between spring and neap tides in the dry season (Figs. 4a-b). However, spring-neap variation in the depth-averaged salinity was less than 1.5 psu in the wet season (Fig. 4c).

3. It is necessary to point out the flow direction when demonstrating the methodology section. Page 7
Answer:
we have added the following sentence in the methodology section in line 178, page 7.
The flow is positive in the upstream direction.

4. The reviewer suggests the authors upgrade the citation (Zhang and Savenije, 2016). e.g., Line 95-96, Page 4; Line 224-226, Page 9; Line 259-260, Page 11; Line 288-289, Line 300-301, Page 12; Line 342-343, Page 14. Also, some ideas (from Savnije, 2005), especially about the empirical coefficient, have been upgraded in the publication (Zhang and Savenije, 2017).

Answer:
We have upgraded the citation in the revised version as per valuable comments.

**Dear Anonymous Referee#2,**

We are very much grateful for your valuable and fruitful comments to improve our manuscript. The referee comment is given in blue font and the answer in black font.

1. What is the relation between $D_i$ (only used in Eq.1) and D and/or $D_t$?
2. What is D, a diffusion coefficient with the dimension $L^2T^{-1}$ as suggested by Eq.(2) and line 200, or a flux as suggested in line 194 (salinity X velocity)?

Answer: D is the longitudinal dispersion with the dimension $L^2T^{-1}$ and $D_t$ is the tide driven dispersion. The subscript i in $D_i(x)$ (Eq. 1 and line 175-176) and $S_i(x)$ or $S_i$ indicates the number of CTD stations where we calculated D using observed salinity $S_i$. As it makes confusion and x can clearly represent the CTD stations, we will delete the subscript i in the revised version to avoid this confusion.